# Neutrophil–lymphocyte ratio across psychiatric diagnoses: a cross-sectional study using electronic health records

Aimee Brinn, James Stone 

Institute of Psychiatry, Psychology and Neuroscience, King's College London, London, UK

**Correspondence to**
Dr James Stone;
james.m.stone@kcl.ac.uk

## ABSTRACT

**Objectives** The main objective of this study was to compare neutrophil–lymphocyte ratio (NLR), a marker of systemic inflammation, between patients diagnosed with International Classification of Diseases 10th Revision (ICD-10) psychiatric disorders and control participants.

**Design** A cross-sectional methodology was employed to retrospectively analyse electronic health records and records derived from a national health survey.

**Setting** A secondary mental healthcare service consisting of four boroughs in South London.

**Participants** A diverse sample of 13 888 psychiatric patients extracted from South London and Maudsley electronic health records database and 3920 control participants extracted from National Health and Nutrition Survey (2015–2016) were included in the study.

**Primary and secondary outcome measures** Primary: NLR levels in patients with mental health diagnoses, NLR between patients with different mental health diagnoses. Secondary: relationship of NLR to length of hospitalisation and to mortality.

**Results** NLR was elevated compared with controls in patients with diagnoses including dementia, alcohol dependence, schizophrenia, bipolar affective disorder, depression, non-phobic anxiety disorders and mild mental retardation (p<0.05). NLR also correlated with age, antipsychotic use and hypnotic use. NLR was found to be higher in individuals of 'white' ethnicity and lower in individuals of 'black' ethnicity. Elevated NLR was associated with increased mortality (β=0.103, p=2.9e−08) but not with hospital admissions or face-to-face contacts.

**Conclusions** Elevated NLR may reflect a transdiagnostic pathological process occurring in a subpopulation of psychiatric patients. NLR may be useful to identify and stratify patients who could benefit from adjunctive anti-inflammatory treatment.

## Strengths and limitations of this study

► Largest study to date of cross-diagnostic neutrophil–lymphocyte measurements in a psychiatric population.
► Sample is representative of diverse adult psychiatric patients in South London.
► Systematic differences between patients and controls reduce the validity of these comparisons.
► This study was retrospective and thus, confounding measures such as body mass index, smoking status and diet were unavailable.

Blood biomarkers are the most commonly used method to study inflammatory processes in psychiatry,[4 5] but many of these are costly or difficult to collect routinely. The neutrophil-to-lymphocyte ratio (NLR), initially developed as a simple method to assess the level of systemic inflammation in critically ill patients,[6] has more recently been employed to assess systemic inflammation in psychiatric patients.[7 8] NLR can be calculated from a full blood count and is thus cheaper and more readily available than cytokine testing.[9] As it constitutes parameters from both innate (neutrophil) and adaptive (lymphocyte) immune systems, it may be less affected by confounding variables, such as exercise, compared with other commonly used inflammatory biomarkers.[9] NLR has demonstrated reliable prognostic value across a range of physical health disorders,[10–13] highlighting a positive association between systemic inflammation and worse clinical outcomes.

The most studied inflammatory biomarkers in psychiatry, interleukin 6 (IL-6), tumour necrosis factor (TNF) and C-reactive protein (CRP), are elevated in patients with schizophrenia, bipolar disorder (BD) and depression, compared with controls.[5] NLR also appears to be elevated in major psychiatric disorders compared with controls.[7 8 14] However, the vast proportion of studies have studied a single disorder in comparison

## INTRODUCTION

Inflammation is a key component of our immune response to physiological injury or infection.[1] Low-grade systemic inflammation is an attenuated, but persistent, form of the inflammatory response and has been found to be prevalent across a range of psychiatric diagnoses, including psychotic, mood, neurotic and personality disorders.[2] It has been suggested that neuroinflammation may underlie the pathology of these conditions.[3]

to healthy controls, or at best, two different psychiatric disorders.[5] True transdiagnostic studies are rare with only three studies to date comparing NLR between psychiatric diagnoses.[15–17]

Of the NLR studies comparing different psychiatric diagnoses, Ozdin *et al* reported that patients with schizophrenia, experiencing a psychotic episode, had higher NLR than patients with manic-phase BD, suggesting that there are detectable differences in NLR across diagnosis and symptomatology.[15] Mazza *et al* found that patients with manic-phase BD had higher NLR than patients with major depressive disorder (MDD).[16] They also found significant differences between manic-phase and depressive-phase BD but not between depressive-phase BD and MDD suggesting that NLR may be elevated in mania but not depression. Lastly, Baykan *et al* found NLR to be significantly elevated in elderly patients with Alzheimer's disease (AD), compared with aged-matched patients with MDD and Parkinson's disease.[17] Shared symptomatology between AD and MDD is problematic for clinicians differentiating between the disorders and the authors suggested that NLR may be an effective diagnostic marker to aid discrimination.

The current evidence base is limited and inconsistent. Further research is required to investigate the role of inflammatory processes in psychiatric illness. Differentially elevated inflammation between different diagnoses may suggest that inflammatory processes are more common in specific disorders. Alternatively, comparable elevation between diagnoses may suggest common aetiological pathways, such as shared immunological genetic vulnerabilities and inflammatory processes. Patients with elevated NLR may represent a subtype of psychiatric illness that does not fit within current diagnostic categories. Lastly, systemic inflammation may be a byproduct of mental health difficulties, reflecting a non-specific comorbid physiological stress.

In this study, we aimed to characterise NLR across a range of International Classification of Diseases 10th Revision (ICD-10) psychiatric diagnoses and to investigate the relationship between NLR and adverse clinical outcomes.

## METHODS
### Design
This study employed a cross-sectional methodology to analyse electronic health records extracted using the Clinical Record Interactive Search (CRIS) tool.[18] We retrospectively analysed neutrophil and lymphocyte counts of psychiatric patients using whole blood count data. For comparison, a control group was obtained from the National Health and Nutrition Examination Survey (NHANES).[19]

### Clinical Record Interactive Search
The CRIS application was developed in 2007, with funding from the British National Institute for Health Research. The application generates a data resource consisting of anonymised electronic health records of service users of the South London and Maudsley (SLaM) National Health Service Foundation Trust.[18] These records are contained in the Biomedical Research Centre (BRC) case register, which is updated every 24 hours. CRIS was used to access patient data, using SQL management studio as a platform for data extraction.

### Patient sample
The SLaM BRC case register includes all patients who have accessed secondary mental healthcare services within a catchment area of four south London boroughs (Lambeth, Southwark, Lewisham and Croydon). SLaM is one of the largest mental healthcare organisations in Europe, providing care to 1.3 million people.[20]

We extracted data from records for all patients who accessed SLaM secondary mental healthcare between 1 January 2007, when CRIS was established, and 20 November 2018, when our data were extracted.

Data were extracted for all patients meeting the following inclusion criteria:
- ≥ 16 years old at earliest recorded blood test.
- ICD-10 Primary Diagnosis F01–F79.
- Recorded neutrophil count.
- Recorded lymphocyte count.

Thirteen thousand eight hundred and eighty-eight patients met our inclusion criteria and were included in the sample.

### Control sample
The control group comprised a sample of participants from the 2015–2016 National Health and Nutrition Survey.[19] The original sample included 9971 participants from across 30 locations in the USA. To enable provision of health status indicators for the varied resident population of the USA, the NHANES oversamples certain population subgroups. In the 2015–2016 survey cycle, the NHANES oversampled the following subgroups: Hispanic persons, non-Hispanic black persons, non-Hispanic Asian persons, persons at or below 185% of the Department of Health and Human Services poverty guidelines,[21] and persons aged 80 years and older.

We included individuals meeting the following inclusion criteria:
- ≥16 years old at time of NHANES (2015–2016) survey.
- Recorded neutrophil count.
- Recorded lymphocyte count.

Office for National Statistics (ONS) guidelines do not code 'Hispanic' as a distinct ethnic group.[22] Hispanic was therefore not a distinct ethnic group in the CRIS database. For ease of sample comparison, we excluded Hispanic participants from the NHANES cohort . Following exclusion, 3920 participants from the NHANES survey met our criteria and were included in the control sample. Any Hispanic participants in the patient group are included by default in the 'Other/Mixed' category.

## Variables

### Demographics

Demographic variables were retrieved from structured fields in the CRIS database and included age at earliest blood test, gender and ethnicity. Ethnicity was coded using ONS guidelines.[22] To facilitate comparison with the NHANES sample, we combined 'Other' and 'Mixed' ethnicity codes into one category 'Other/Mixed'.

### Neutrophil-to-lymphocyte ratio

Absolute neutrophil and lymphocyte counts were obtained from routine blood testing data, attained at the point of patient entry to SLaM secondary care services. NLR was calculated for each patient by dividing neutrophil count by lymphocyte count. We extracted earliest available blood count values to minimise potential confounding effects of secondary healthcare treatment.

### ICD-10 diagnosis

We included diagnoses from eight ICD-10 diagnostic blocks in this study: organic, including symptomatic, mental disorders (F01–F09), mental and behavioural disorders due to psychoactive substance use (F10–F19), schizophrenia, schizotypal and delusional disorders (F20–F29), mood (affective) disorders (F30–F39), neurotic, stress-related and somatoform disorders (F40–F49), behavioural syndromes associated with physiological disturbances and physical factors (F50–59), disorders of adult personality and behaviour (F60–F69) and mental retardation (F70–F79). Diagnostic variables were defined by the primary diagnosis assigned to the patient closest to their blood test result.

We excluded patients with diagnoses in the disorders of psychological development (F80–F89) and behavioural/emotional disorders with onset usually occurring in childhood and adolescence (F90–F98) diagnostic blocks as most patients were under 16 years old. We also excluded patients in the unspecified mental disorder (F99) block due to ambiguity of the diagnosis.

### Adverse clinical outcomes

We defined 'number of overnight bed stays' as the total number of overnight hospital stays for each patient since the date of the blood test. Face-to-face events are community face-to-face contacts between a patient and a SLaM-affiliated health professional. This includes patient contact with Community Mental Health Teams, Home Treatment Teams and Liaison A&E; it does not include inpatient contacts. A patient may have multiple face-to-face events per day. The total number of 'face-to-face events' for each patient is measured since the date of the blood test.

Both measures were adjusted for patient time in services since the date of the blood test. Death was recorded as a dichotomous mortality variable.

### Medication

The medication prescribed to the patient within 1 year of the blood test date was extracted. Medications were

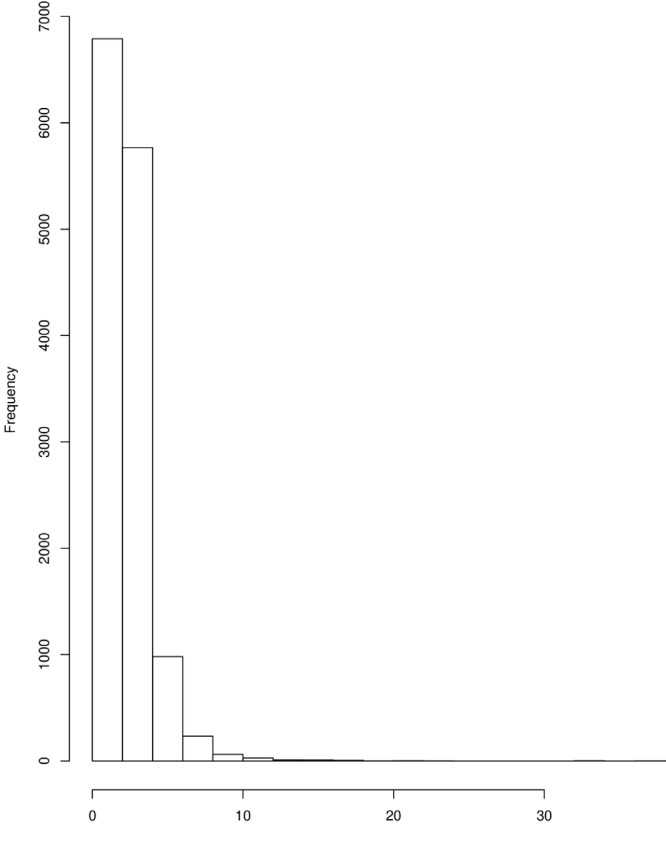

**Figure 1** Distribution of neutrophil-to-lymphocyte ratio (NLR) in patient sample.

categorised into four classes: antipsychotics, antidepressants, mood stabilisers and hypnotics and were analysed as binary variables.

### Statistical analysis

We conducted all statistical analyses for this study using R (V.3.6). We analysed data distribution by use of histogram. We report the mean and SD for normally distributed data.

Because NLR data demonstrated an exponential distribution (see figures 1 and 2), we conducted regression analyses using a log-linked gamma-family generalised linear model. For each analysis, the shape parameter of the model was estimated using the gamma.shape function from the MASS package, and this was used to set the dispersion parameter as 1/(maximum likelihood estimate). We ran the following models: (1) control NLR as dependent variable versus NLR in ICD-10 diagnoses; (2) NLR between all ICD-10 diagnoses that were significantly different from control NLR from model 1. In models 1 and 2, age, ethnicity, gender and medication were included as independent variables. We then ran three separate models within the CRIS dataset only to investigate the association of NLR with clinical outcome measures. These models set NLR as the dependent variable versus death, hospital admissions or numbers of face-to face contact. ICD-10 diagnosis, age, ethnicity, gender and medication were included as independent variables. We also ran each

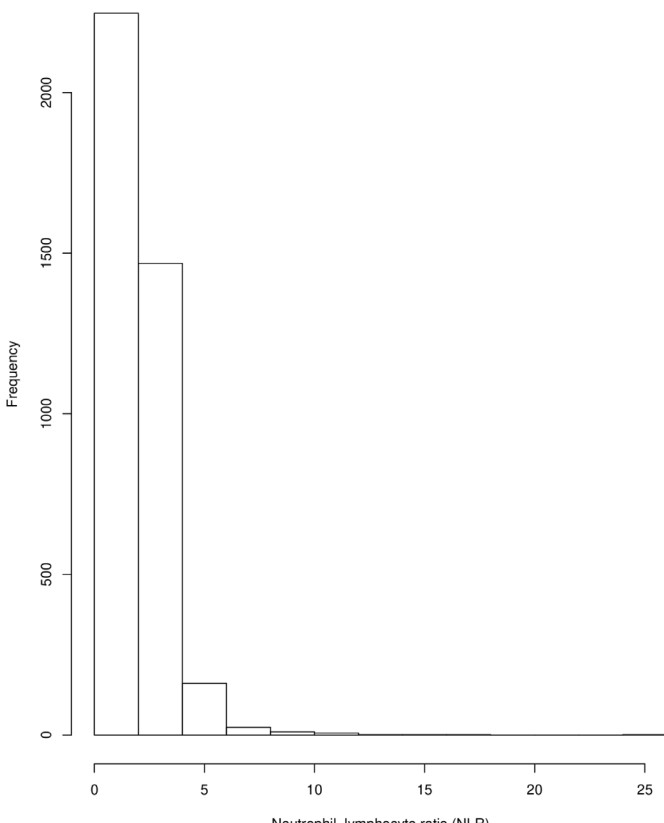

**Figure 2** Distribution of neutrophil-to-lymphocyte ratio (NLR) in control sample.

**Table 1** Distribution of age, ethnicity and gender variables across patient and control samples

| | Patient sample | | Control sample | |
|---|---|---|---|---|
| | **N** | **%** | **N** | **%** |
| **Age group** | | | | |
| 16–19 | 1201 | 8.65 | 350 | 8.93 |
| 20–29 | 3188 | 22.95 | 600 | 15.31 |
| 30–39 | 2989 | 21.52 | 591 | 15.08 |
| 40–49 | 2759 | 19.87 | 619 | 15.79 |
| 50–59 | 1987 | 14.31 | 584 | 14.90 |
| 60–69 | 878 | 6.32 | 537 | 13.70 |
| 70–79 | 561 | 4.04 | 366 | 9.34 |
| 80+ | 325 | 2.34 | 273 | 6.96 |
| **Ethnicity** | | | | |
| Asian | 788 | 5.67 | 668 | 17.04 |
| Black | 3749 | 26.99 | 1187 | 30.28 |
| Other/mixed | 1238 | 8.91 | 217 | 5.54 |
| White | 8113 | 58.43 | 1848 | 47.17 |
| **Gender** | | | | |
| Male | 6539 | 47.08 | 1945 | 49.6 |
| Female | 7346 | 52.9 | 1975 | 50.4 |
| Non-specified | 3 | 0.02 | – | – |

n=number in group, %=relative percentage of total sample.

model unadjusted for covariates, as recommended by the Strengthening the Reporting of Observational Studies in Epidemiology statement.[23]

## RESULTS
### Demographic results
The mean ages of patients and controls were 40.1 (SD: 16.57) and 46.9 (SD: 19.47), respectively. Due to large sample size and consequent high power of the study to detect small differences, variables were not tested for differences in distribution. Distribution of demographic variables for each group are displayed in table 1. There was no relationship between date of blood test (from which the NLR value was derived) and NLR value across the patient sample ($\beta$=−4.76e−06, p=0.692) suggesting that NLR measurement remained stable across the defined time period (2007–2018).

### Model 1: NLR between controls and patients
NLR in the following ICD-10 diagnoses was significantly higher than controls: F00, AD ($\beta$=0.15, p=1.3e−06); F01, vascular dementia ($\beta$=0.24, p=0.003); F02 dementia in other diseases ($\beta$=0.27, p=0.008); F05, delirium ($\beta$=0.28, p=0.003); F06, other mental disorders due to brain damage and dysfunction ($\beta$=0.16, p=0.0006); F10, mental and behavioural disorders due to alcohol ($\beta$=0.16, p=2.0e−14); F20, schizophrenia ($\beta$=0.09, p=9.7e−08); F23, acute and transient psychotic disorders ($\beta$=0.16, p=3.7e−08); F25,

schizoaffective disorders ($\beta$=0.07, p=0.002); F28, other non-organic psychotic disorders ($\beta$=0.17, p=0.03); F29, unspecified non-organic psychotic disorders ($\beta$=0.13, p=1.9e−07); F30, manic episode ($\beta$=0.20, p=3.0e−06); F31, bipolar affective disorder ($\beta$=0.12, p=3.0e−07); F32, depressive episode ($\beta$=0.13, p=7.31e−11); F33, recurrent depressive disorder (0.14, p=7.9e−07); F38 other mood (affective) disorders ($\beta$=0.41, p=0.001); F41, other (non-phobic) anxiety disorders ($\beta$=0.06, p=0.03); F43, reaction to severe stress, and adjustment disorders ($\beta$=0.08, p=0.004); F60, specific personality disorders ($\beta$=0.09, p=0.0005); F61, mixed personality disorders ($\beta$=0.18, p=0.03); F70, mild mental retardation ($\beta$=0.14, p=0.02) and F79, unspecified mental retardation ($\beta$=0.40, p=0.047). NLR in all other diagnoses was not significantly different from controls. In the model, age was positively correlated with NLR ($\beta$=0.005, p<2e−16; figure 3). Black ethnicity was associated with lower NLR ($\beta$=−0.23, p<2e−16) and white ethnicity with higher NLR ($\beta$=0.09, p=5.8e−07; figure 4). Antipsychotic use ($\beta$=0.04, p=0.003) and hypnotic use ($\beta$=0.03, p=0.01) were both associated with higher NLR (see online supplementary materials table T1). A secondary analysis unadjusted for covariates revealed a similar pattern of NLR elevation across diagnostic groups compared with controls (see online supplementary materials table T2).

### Model 2: NLR between diagnoses with elevated NLR
We found that of the diagnoses elevated in model 1, patients with a diagnosis of F38, other mood (affective)

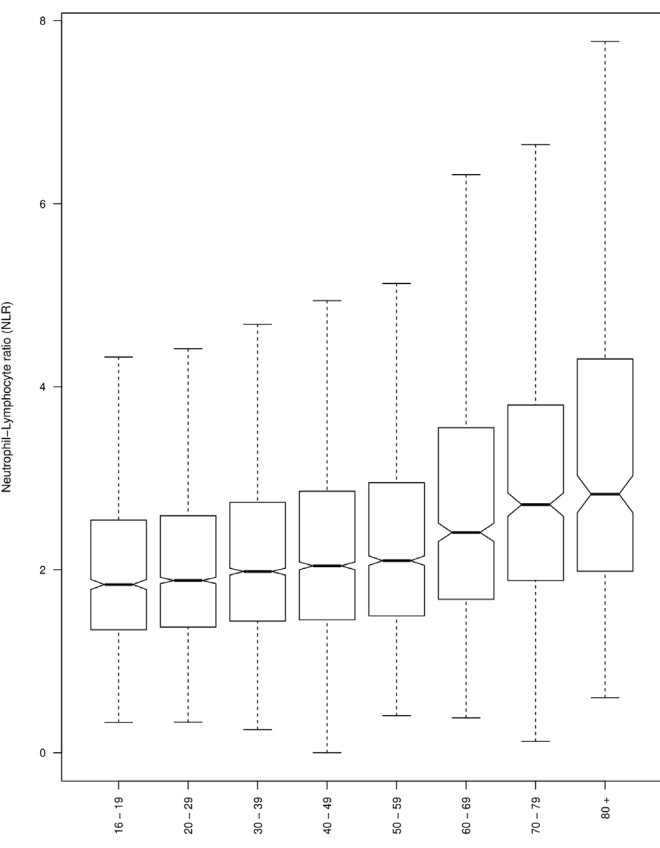

**Figure 3** Neutrophil-to-lymphocyte ratio (NLR) (uncorrected) across ethnicity.

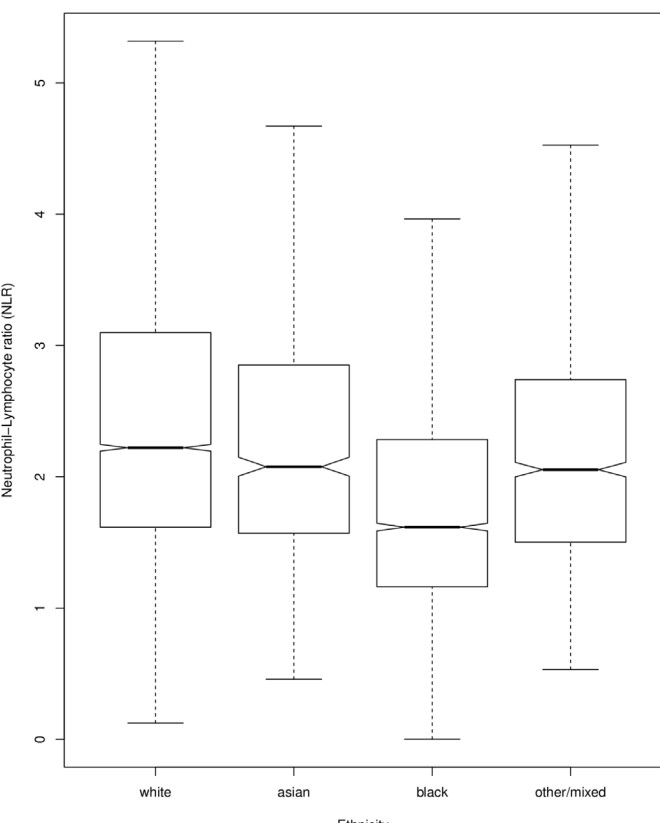

**Figure 4** Neutrophil-to-lymphocyte ratio (NLR) (uncorrected) across age groups.

disorders, had higher NLR (β=0.27, p=0.04), whereas all other diagnoses did not differ significantly (see online supplementary materials table T3). When excluding covariates from the model, we found that many of the diagnostic categories had lower NLR than AD (see online supplementary materials table T4) but given the strong relationship between age and ethnicity on NLR, the significance of this finding is uncertain.

### Model 3: NLR across adverse clinical outcomes

There was a positive association between mortality and NLR, both unadjusted (β=0.274, p=2e-16) and after correcting for ICD-10 diagnostic group, age, gender, ethnicity, antipsychotic, antidepressant, mood stabiliser and hypnotic medication use (β=0.103, p=2.9e−08; see online supplementary materials table T5).

There was no significant relationship between number of overnight bed stays or number of face-to-face events and NLR.

### DISCUSSION

This study suggests that systemic inflammation occurs in several psychiatric diagnoses. In particular, NLR elevation was seen in patients with dementia, alcohol dependence, schizophrenia, bipolar affective disorder, depression, non-phobic anxiety disorders, specific personality disorders and mild mental retardation. In those groups with

elevated NLR, the only diagnostic group that had significantly higher NLR was F38 'other mood disorders'. All other diagnoses with elevated NLR did not significantly differ in terms of NLR level, suggesting that NLR elevation is a non-specific, cross-diagnostic finding. It is possible that patients with elevated NLR have a shared inflammatory aetiology to their different clinical presentations. In contrast, certain diagnoses, including substance misuse (excluding alcohol), eating disorders and mental retardation (excluding mild mental retardation), did not show significantly elevated NLR from controls after correcting for covariates, suggesting that an inflammatory aetiology is not present in all diagnoses.

The Research Domain Criteria (RDoC) approach reconceptualises psychopathology into transdiagnostic domains,[24] presenting a novel classification of mental illnesses, based on dimensions of observable behaviour and neurobiological measures.[25] As our results indicate that inflammatory processes are shared across diagnoses, inflammation may play a role in symptom domains shared by these conditions. For example, alterations in serotonergic, dopaminergic and glutamatergic pathways are consistently reported across mood,[26–28] and psychotic disorders,[29 30] and have been reported to be mediated by immunological pathways.[31–33] Immune-mediated alterations in serotoninergic and glutamatergic systems may result in mood and cognitive symptoms while inflammation-induced modifications of dopaminergic

systems may result in motivational deficits found across different psychiatric disorders.[34]

In contrast to previous reports,[15–17] we did not find NLR to differ between bipolar affective disorder and schizophrenia, between bipolar affective disorder or depression, or between depression and AD after correcting for covariates. This contradiction in findings may be attributed to the larger sample size in our study. Alternatively, NLR may change with acute symptomatology, a hypothesis we were not able to directly test with our dataset.

Our results align closely with findings from other inflammatory biomarkers. A recent meta-analysis indicated that variation patterns in inflammatory biomarkers IL-6, TNF and CRP were comparable across diagnoses.[5] Thus, current evidence suggests that phenomenologically defined diagnostic criteria do not map closely onto biologically based pathogenic markers such as NLR.[35]

Our study also replicated the finding that a positive association exists between NLR and mortality.[36] However, we did not find any relationship between NLR and patient outcomes; there was no relationship between NLR and 'number of overnight bed stays' or 'number of face-to-face events'. This is in contrast to a previous study which found a positive relationship between number of hospitalisations and NLR in patients with BD.[37] The study involved a small sample of 80 patients, but the findings were strengthened by the fact the authors employed a longitudinal design. The difference in findings with our study could be due to the cross-diagnostic nature of our study, and the fact that the patients in our study were not all hospitalised.

A positive relationship between mortality and NLR has been indicated across physical health disorders.[10 12 38] As we were not able to adjust for physical health comorbidities, the relationship that we found may reflect an association between systemic inflammation and poor physical health. Alternatively, our findings may reflect an independent positive relationship between NLR and mortality in psychiatric patients. In both patients with BD and patients with MDD, NLR has been found to be a significant predictor of suicide risk,[39 40] which may account for some of the increased mortality. However, elevated NLR has been found to independently predict mortality, irrespective of age, sex, education, metastatic cancer, liver disease and depression,[36] suggesting that systemic inflammation may reduce survival through alternative pathological pathways.

## Limitations

This study has several limitations. Confounding factors, which may modulate NLR such as body mass index,[41] smoking,[42] diet and exercise,[43] were not controlled for and so interactions such as a positive correlation between depression and obesity[44] may have confounded our results. Likewise, comorbid physical conditions found to modulate NLR, such as cancer or autoimmune disease,[38 45 46] or any concurrent infection,[47] were neither excluded nor adjusted for. Moreover, although medications were adjusted for, dosage was not accounted for and this may confound our results. Additionally, only four classes of psychotropic drugs were included in our model; other medications, such as steroid and non-steroidal anti-inflammatory drugs, are known to affect systemic inflammation but were neither excluded nor adjusted for.

The control sample represents a non-institutionalised civilian (US) population. These participants were not screened for physical or mental disorders, so the prevalence of these conditions in the control sample is unknown. The fact that samples were selected from different countries and blood cell essays were performed in different laboratories reduces the validity of our patient and control comparisons. However, although NLR was elevated in some ICD-10 diagnoses compared with the control group, NLR in patients with other diagnoses did not significantly differ from this control sample, suggesting that the control NLR was suitable as a comparison group in this study.

This study did not employ validated symptom severity measures. 'Number of overnight bed stays' and 'number of face-to-face events' are logical outcome measures of illness severity. These real-world measures increased the clinical transferability of our results, but they may be affected by unknown factors, such as help-seeking attitudes or illness insight.

This sample is representative of a population seeking secondary care services. Results may not be generalisable to patients seeking primary care services. Finally, this is a cross-sectional study based on a single measurement of NLR for each patient. The one measurement may not reflect inflammatory status over time and causal inferences cannot be made.

Nonetheless, this is the largest study to date of cross-diagnostic NLR measurements in a psychiatric population and our sample is representative of adult psychiatric patients in South London. Every patient who accessed SLaM services within our time frame and had available blood count data from age 16 years old or above was included in our sample. This minimised selection biases. Previous studies included only elderly,[17] or Caucasian patients,[16] and so were less generalisable. Our sample is diverse in age and ethnicity and so demonstrates improved translational potential. This study also compares NLR across the broadest range of diagnoses to date and employs the largest sample yet to address these two research aims.

## Future directions

Findings suggest broad transdiagnostic commonalities regarding systemic inflammation. As within-diagnostic differences in inflammation have also been found,[48 49] it appears that inflammatory processes are not bound to traditional diagnostic categories. In both depression and schizophrenia, inflammation is thought to contribute to treatment resistance,[50 51] perhaps contributing to poor primary treatment response rates in psychiatry.[52 53]

It may be fruitful for future studies investigating NLR in psychiatric populations to employ an RDoC approach.[24] NLR may be a cost-effective and efficient biomarker by which to stratify psychiatric patients on a common pathophysiological basis. Conducting research without the restraints of nosology may elucidate transdiagnostic symptom domains associated with systemic inflammation. Recent years have seen pharmaceutical companies withdraw investments from psychiatry due to 'low probability of success',[54] and nosology has been highlighted as one of the barriers to research progress.[55] Transdiagnostic biological abnormalities may constitute common targets for pharmacological treatment and NLR may be an accessible tool to aid in identification and stratification of these patients in clinical trials.

Lastly, future research should aim to measure NLR levels prior to and following successful treatment; changes in NLR post treatment may indicate that increased NLR levels are associated with acute symptomatology, rather than inherent vulnerability.

## CONCLUSIONS

Systemic inflammation may occur as a transdiagnostic pathological process in a subpopulation of psychiatric patients. NLR may be an effective biomarker to identify these patients who may benefit from adjunctive anti-inflammatory pharmacological treatment.

**Contributors** The study was proposed by JS. This proposal provided an outline of study objectives, methods and statistical analyses. Data extraction was specified by JS and completed by AB. Supervised by JS, AB conducted a literature review which informed further development of the study design, such as indicating which confounding variables were to be included in the regression model. AB completed the abstract, introduction, methods, data analysis, results and discussion for the first draft of the manuscript. JS provided guidance throughout with specific instruction for the data analysis and results sections. Both authors contributed to the final manuscript and approved the final version.

**Funding** This paper represents independent research funded by the National Institute for Health Research Biomedical Research Centre at South London and Maudsley NHS Foundation Trust and King's College London.

**Disclaimer** The views expressed are those of the authors and not necessarily those of the NHS, the NIHR or the Department of Health and Social Care.

**Competing interests** None declared.

**Patient and public involvement** Patients and/or the public were not involved in the design, or conduct, or reporting, or dissemination plans of this research.

**Patient consent for publication** Not required.

**Ethics approval** The CRIS data resource received ethical approval as an anonymised dataset for secondary analyses from Oxford Research Ethics Committee C (18/SC/0372).

**Provenance and peer review** Not commissioned; externally peer reviewed.

**Data availability statement** No data are available. Because of their clinical source and under the terms of Ethics and Information Governance approvals, datasets derived from CRIS have to remain within the SLaM firewall. However, all data used for this study will be made accessible on request to cris.administrator@slam.nhs.uk subject to setting up an appropriate SLaM honorary contract or research passport.

**ORCID iD**
James Stone http://orcid.org/0000-0003-3051-0135

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
