## [Reviewer comments · BMJ Open]

ARTICLE DETAILS

TITLE (PROVISIONAL)	Neutrophil-lymphocyte ratio across psychiatric diagnoses: A cross-sectional study using electronic health records
AUTHORS	Brinn, Aimee; Stone, James

VERSION 1 – REVIEW

REVIEWER	Mario Gennaro Mazza University Vita-Salute San Raffaele, Milan, Italy
REVIEW RETURNED	11-Feb-2020

GENERAL COMMENTS	The authors propose a retrospective study regarding the values of Neutrophil/lymphocyte ratio (NLR), across large psychiatric diagnoses using an electronic health record investigation. The area of research is quite new and there is an increasing amount of studies that investigate this cheap and easy available marker in mental health. I recommend the publication of the manuscript following minor revision. Page 9, line 13: "Hispanic, Black and Asian individuals are overrepresented in the sample, as are individuals at or below 185 percent of the Department of Health and Human Services poverty guidelines (Burwell, 2015) and those aged 80 and over". This sentence is misleading. Page 9, lines 34-45: "Office for National Statistics (ONS) (Office for National Statistics, 2009) guidelines do not code 'Hispanic' as a distinct ethnic group. 'Hispanic' was therefore not a distinct ethnic group in the CRIS database. For ease of sample comparison, we excluded Hispanic participants from our study. 3,920 participants met our criteria and were included in the control sample". It is not clear if Hispanics were excluded both from control and patients groups. Page 11, line 18: "We defined 'number of face-to-face events' as the number of clinical face-to-face contacts, such as hospital appointments, for each patient since the date of the blood test". Considering that face-to-face events is indirect and not validated symptom severity measures (as stated by the authors in the limitation section) an extensive description of this variable is needed. Please specify which kind of face-to-face events were considered apart from hospital appointments (outpatient clinic evaluation, emergency department entry etc.).
--

	Page 30, paragraph medication: the authors described the medication status only as a binary variable (antidepressant yes/no, mood stabilizer yes/no etc.), and not as dose or medication load for each medication class (imipramine equivalents, chlorpromazine equivalents etc.). This should be considered a limitation of the study. Moreover, several medications (steroid and NSAIDs drugs) are known to affect the NLR and inflammatory status in general but the authors did not exclude nor adjusted for this confounding. Page 12, Demographic Results: were the controls and patients group comparable for age, sex and ethnicity, or they differ? Page 15, lines 3-18: “In comparing our findings to published data, one previous study of NLR found that patients with acute schizophrenia had higher NLR than patients with manic phase BD (Ozdin et al, 2017), whereas Mazza and colleagues (2019b) did not find any difference in NLR between patients with bipolar and unipolar depression. Another group reported that NLR was elevated in patients with Alzheimer’s disease compared to age-matched patients with MDD and Parkinson’s disease (Baykan et al. 2018). We did not find any evidence for elevated NLR in Alzheimer’s disease compared to other diagnoses, but we were not able to exclude the possibility that NLR elevations may occur during acute exacerbations of psychosis”. This paragraph is a repetition (see paragraph on page 6, lines 20-47 in the introduction). I suggest to the authors synthesize the previous finding in the introduction and to deeper and extensively discuss it in detail in the discussion. Page 13, line 14: “In terms of the association of NLR with patient outcomes, we found a significant positive relationship between NLR and mortality” This sentence is misleading.
--	---

REVIEWER	Kaijun Niu Nutritional Epidemiology Institute, Tianjin Medical University, Tianjin, China
REVIEW RETURNED	30-Mar-2020

GENERAL COMMENTS	 1. It may be worthwhile to see the levels in these patients once they are treated for psychiatric disorders so as to see if the NLR levels change with treatment. 2. Another confounder is the NLR levels will fluctuate based on any concurrent infections/malignancy/atherosclerosis/arthritis/COPD in these patients, and I am unsure if this was accounted for. 3. Speculations should be given in the discussion why there is association for NLR with psychiatric disorders. 4. This study is based on single measurements of NLR, which may not reflect inflammatory status over time. 5. Please narrate detailedly what is new about NLR and how it is a predominant inflammatory marker.
---

VERSION 1 – AUTHOR RESPONSE

Reviewer 1

1. Page 9, line 13: "Hispanic, Black and Asian individuals are overrepresented in the sample, as are individuals at or below 185 percent of the Department of Health and Human Services poverty guidelines (Burwell, 2015) and those aged 80 and over". This sentence is misleading

Thank you for this comment. We have now amended this sentence as follows:

"To enable provision of health status indicators for the varied resident population of the United States, the NHANES survey oversamples certain population subgroups. In the 2015-2016 survey cycle, the NHANES oversampled the following subgroups: Hispanic persons, Non-Hispanic black persons, Non-Hispanic Asian persons, persons at or below 185 percent of the Department of Health and Human Services poverty guidelines (Burwell, 2015) and persons aged 80 years and older."

2. Page 9, lines 34-45: "Office for National Statistics (ONS) (Office for National Statistics, 2009) guidelines do not code 'Hispanic' as a distinct ethnic group. 'Hispanic' was therefore not a distinct ethnic group in the CRIS database. For ease of sample comparison, we excluded Hispanic participants from our study. 3,920 participants met our criteria and were included in the control sample".

It is not clear if Hispanics were excluded both from control and patients groups.

We have now addressed this point in the text: "Any Hispanic participants in the patient group are included by default in the 'Other/Mixed' category."

3. Page 11, line 18: "We defined 'number of face-to-face events' as the number of clinical face-to-face contacts, such as hospital appointments, for each patient since the date of the blood test". Considering that face-to-face events is indirect and not validated symptom severity measures (as stated by the authors in the limitation section) an extensive description of this variable is needed. Please specify which kind of face-to-face events were considered apart from hospital appointments (outpatient clinic evaluation, emergency department entry etc.).

We have expanded on this point as requested: "Face-to-face events' are community face-to-face contacts between a patient and a SLAM-affiliated health professional. This includes patient contact with Community Mental Health Teams, Home Treatment Teams and Liaison A&E; it does not include inpatient contacts. A patient may have multiple face-to-face events per day. The total number of 'face-to-face events' for each patient is measured since the date of the blood test."

4. Page 30, paragraph medication: the authors described the medication status only as a binary variable (antidepressant yes/no, mood stabilizer yes/no etc.), and not as dose or medication load for each medication class (imipramine equivalents, chlorpromazine equivalents etc.). This should be considered a limitation of the study.
Moreover, several medications (steroid and NSAIDs drugs) are known to affect the NLR and inflammatory status in general but the authors did not exclude nor adjusted for this confounding.

We have addressed this point as follows: "Moreover, although medications were adjusted for, dosage was not accounted for and this may confound our results. Additionally, only four classes of psychotropic drugs were included in our model; other medications, such as steroid and NSAID drugs, are known to affect systemic inflammation but were neither excluded nor adjusted for."

5. Page 12, Demographic Results: were the controls and patients group comparable for age, sex and ethnicity, or they differ?

We have added the following: “Due to large sample size and consequent high power of the study to detect small differences, variables were not tested for differences in distribution. Distribution of demographic variables for each group are displayed in Table 1.”

6. Page 15, lines 3-18: “In comparing our findings to published data, one previous study of NLR found that patients with acute schizophrenia had higher NLR than patients with manic phase BD (Ozdin et al, 2017), whereas Mazza and colleagues (2019b) did not find any difference in NLR between patients with bipolar and unipolar depression. Another group reported that NLR was elevated in patients with Alzheimer’s disease compared to age-matched patients with MDD and Parkinson’s disease (Baykan et al. 2018). We did not find any evidence for elevated NLR in Alzheimer’s disease compared to other diagnoses, but we were not able to exclude the possibility that NLR elevations may occur during acute exacerbations of psychosis”. This paragraph is a repetition (see paragraph on page 6, lines 20-47 in the introduction). I suggest to the authors synthesize the previous finding in the introduction and to deeper and extensively discuss it in detail in the discussion

This is helpful. We have removed the repetitive paragraph and added the following to the discussion: “In contrast to previous reports (Baykan et al. 2018, Ozdin et al. 2017, Mazza et al. 2019), we did not find NLR to differ between diagnoses. This contradiction in findings may be attributed to the larger sample size in our study. Alternatively, NLR may change with acute symptomatology, a hypothesis we were not able to directly test with our dataset.”

7. Page 13, line 14: “In terms of the association of NLR with patient outcomes, we found a significant positive relationship between NLR and mortality”
This sentence is misleading.

We have amended this sentence as follows: “Furthermore, our study replicated the finding that elevated NLR was associated with an increased risk of mortality (Isaac et al, 2016). However, we did not find any relationship between NLR and patient outcomes; there was no relationship between NLR and ‘number of overnight bed stays’ or ‘number of face-to-face events’.”

Reviewer 2

1. It may be worthwhile to see the levels in these patients once they are treated for psychiatric disorders so as to see if the NLR levels change with treatment.

Thanks for this comment. We agree with this, but unfortunately this was not possible with the current dataset. We have added this to the discussion: “Lastly, future research should aim to measure NLR levels prior to and following successful treatment; changes in NLR post-treatment may indicate that increased NLR levels are associated with acute symptomatology, rather than inherent vulnerability.”

2. Another confounder is the NLR levels will fluctuate based on any concurrent infections/malignancy/atherosclerosis/arthritis/COPD in these patients, and I am unsure if this was accounted for.

Thank you for this comment. This has now been added to the limitations:

“Likewise, comorbid physical conditions found to modulate NLR, such as cancer or autoimmune disease (Bhat et al, 2013, Celikbilek et al, 2013, Templeton et al, 2014), or any concurrent infection (Kim, Jung & Suh, 2017) (Reviewer 2, point 2), were neither excluded nor adjusted for. Moreover, although medications were adjusted for, dosage was not accounted for and this may confound our results. Additionally, only four classes of psychotropic drugs were included in our model; other medications, such as steroid and NSAID drugs, are known to affect systemic inflammation but were neither excluded nor adjusted for.”

3. Speculations should be given in the discussion why there is association for NLR with psychiatric disorders.

We have added the following to the introduction:

“Low-grade systemic inflammation is an attenuated, but persistent, form of the inflammatory response and has been found to be prevalent across a range of psychiatric diagnoses, including psychotic, mood, neurotic, and personality, disorders (Osimo et al, 2018). It has been suggested that neuroinflammation may underlie the pathology of these conditions (Najjar et al. 2013).”

Later in the introduction we highlight why NLR is thought to be an inflammatory marker (see point 5 below).

4. This study is based on single measurements of NLR, which may not reflect inflammatory status over time.

We have now noted this in the text: “This sample is representative of a population seeking secondary care services. Results may not be generalisable to patients seeking primary care services. Finally, this is a cross-sectional study based on a single measurement of NLR for each patient. The one measurement may not reflect inflammatory status over time and causal inferences cannot be made.”

5. Please narrate detailedly what is new about NLR and how it is a predominant inflammatory marker.

We have added the following to the introduction:

“The neutrophil to lymphocyte ratio (NLR), initially developed as a simple method to assess the level of systemic inflammation in critically ill patients (Zahorec, 2001), has more recently been employed to assess systemic inflammation in psychiatric patients (Karageorgiou et al, 2019, Mazza et al, 2018). NLR can be calculated from a full blood count and is thus cheaper and more readily available than cytokine testing (Gibson et al, 2007). As it constitutes parameters from both innate (neutrophil) and adaptive (lymphocyte) immune systems, it may be less affected by confounding variables, such as exercise, compared to other commonly used inflammatory biomarkers (Gibson et al, 2007). NLR has demonstrated reliable prognostic value across a range of physical health disorders (Afari & Bhat, 2016, Chandrashekara et al, 2017, Koh et al, 2015, Wang et al, 2017, highlighting a positive association between systemic inflammation and worse clinical outcomes.”